# Barriers and Facilitators in Perioperative Antibiotic Prophylaxis: A Mixed-Methods Study in a Small Island Setting

**DOI:** 10.3390/antibiotics10040462

**Published:** 2021-04-19

**Authors:** Liza A. M. van Mun, Sabien J. E. Bosman, Jessica de Vocht, Jaclyn de Kort, Jeroen Schouten

**Affiliations:** 1Radboud Center for Infectious Diseases (RCI), RadboudUMC, Geert Grooteplein Zuid 10, 6525 GA Nijmegen, Gelderland, The Netherlands; Jessica.deVocht@radboudumc.nl (J.d.V.); Jeroen.Schouten@radboudumc.nl (J.S.); 2Department of Internal Medicine, Dr. Horacio E. Oduber Hospital, Dr. Horacio E. Oduber Boulevard #1, Oranjestad, Aruba; J.dekort@hoharuba.com

**Keywords:** perioperative antibiotic prophylaxis, small island setting, antibiotic stewardship

## Abstract

Few studies have addressed antibiotic guideline adherence in small island settings, such as Aruba. This study aimed to evaluate the appropriateness of perioperative antibiotic prophylaxis (PAP) and identify barriers for PAP guideline adherence. A mixed-methods study was carried out at the operating theatre (OT) in the Dr. Horacio E. Oduber Hospital (HOH) in Aruba. First, a prospective audit was performed on the appropriateness of guideline-derived quality indicators (QIs). Then, interviews based on the Flottorp framework were conducted to identify barriers for guideline adherence. Finally, a survey was distributed to verify the outcomes of the interviews. The appropriateness of QIs was measured: correct indication (50.6%); antimicrobial agent (30.8%); dose (94.4%); timing (55.0%); route of administration (100%); duration (89.5%); and redosing (95.7%). The overall appropriateness was 34.9%. The main barriers discovered were poor knowledge about PAP and the guidelines and professional interactions regarding PAP, specifically poor communication and lack of clarity about responsibilities regarding PAP. This study was the first to evaluate the appropriateness and to identify barriers for PAP guideline adherence in a small island hospital. The overall appropriateness of PAP was poor with just 34.9%. Future interventions should be focused on communication, education and awareness of the possibility to consult an ID physician or microbiologist.

## 1. Introduction

Surgical site infections (SSIs) are common surgical complications and continue to be a leading component of nosocomial morbidity and mortality [1]. SSI rates vary between 3 and 20%, dependent on the operative procedure and risk factors [2], and are associated with increased treatment costs, prolonged hospital stay and increased mortality [3]. For example, an SSIs post-Caesarean section cause high maternal mortality in Sierra Leone [4]. Using perioperative antibiotic prophylaxis (PAP) significantly decreased the risk of SSIs with 13–39% [5]. Large variation exists in the appropriate use of PAP between countries worldwide, with percentages of appropriateness varying between 0.3 and 84.5% [6].

The inappropriate use of PAP may reduce treatment efficacy and induce antimicrobial resistance (AMR) [7]. The World Health Organisation (WHO) declares AMR as a global health and development threat. AMR leads to high healthcare costs, less effective treatment for infections and less success of major surgeries [8]. To delay the process of AMR, antibiotic stewardship programs (ASPs) are developed to improve appropriate antibiotic use and monitor AMR [9]. The primary goal of ASPs is to improve antibiotic utilization and thus optimise patient outcomes, with a secondary goal of cost reduction [10]. According to the WHO ASP guidelines, PAP should be included in these programs [11].

Health professionals in Aruba struggle with high AMR rates compared to other countries [12]. Aruba is a small island developing state (SIDS) in the Caribbean. The WHO recognizes a group of 58 island nations across different geographical divisions—Caribbean, Pacific, Atlantic, Indian Ocean, Mediterranean and South China Sea as Small Island Developing States. Among many challenges, due to their contained nature, the potential impact of infectious diseases in SIDSs on its population is severe [13]. Combatting antimicrobial resistance and the significance of functioning ASPs is of great importance. Few data have been published on antibiotic stewardship in SIDSs.

Several studies are being conducted on the appropriateness of antibiotic use in the Dr. Horacio E. Oduber Hospital (HOH) [12,14]. However, PAP was not included in these studies and data on PAP guideline adherence in the HOH are lacking. Therefore, overall adherence rates and reasons for noncompliance to PAP guidelines remain unclear. Non-adherence could be affected by barriers such as patient factors, system factors, physician factors and cultural factors [15].

This study aimed to evaluate the appropriateness of PAP through a mixed-methods approach by first performing a prospective audit and subsequently conducting interviews and distributing a survey to identify barriers for PAP guideline (non-)adherence. The outcomes of this study can be advantageous to improve ASPs in Aruba and other SIDS.

## 2. Results

### 2.1. Audit

Eighty-three patients undergoing surgery were enrolled in the audit. Table 1 summarises patients’ characteristics and surgical information. The mean age was 54.1 years (range 18–89) and 57% were female. Most procedures were elective (74%). The major types of surgery were general surgery (27.7%), gynaecology (22.9%), neurosurgery (12%), and orthopaedic surgery (12%). In 11 procedures an implant was used, of which six were cardiac devices and five were orthopaedic prosthetics. In none of the cases was an ID physician consulted.

Table 2 shows the appropriateness of PAP use per QI. Of all 83 patients, 26 patients correctly did not receive PAP and three received appropriate prophylaxis based on all QIs (34.9%).

Indication: In 41 patients, indication for prophylaxis was non-compliant to the guidelines (49.4%): 23 people received perioperative prophylaxis when this was not indicated, and two people did not receive prophylaxis while this was indicated. The remaining 16 patients who received inappropriate prophylaxis had indication for a combination of two agents, namely patients with certain abdominal or gynaecological procedures; however, three received none and 13 received only one agent.

Antimicrobial agent: Of all 52 patients who received PAP, 16 received the appropriate agent (30.8%). Twenty-three patients received an agent, even though PAP was not indicated. Thirteen patients should have received two agents, but only received one. All three patients with an allergy to penicillin received an appropriate substitute agent.

Dose: The dose was appropriate in 51 of 54 cases (94.4%); the three incorrect dosages (too low) were caused by not adjusting the dose according to weight.

Timing: The timing of prophylaxis was correct in 22 of 40 cases (55.0%), based on the researchers’ direct observations: the antibiotic was given too late in 15 cases and too early in three cases. When based on the incision and administration times noted on the operating theatre (OT) forms, timing was appropriate in 21 of 49 cases (42.9%). In 24 cases, the antibiotic was given too late and in four cases too early. In the OT forms of 11 patients, no PAP was reported, even though the administration of PAP was observed by the researchers.

Duration: Fifty-one patients received correct additional PAP within 24 h after surgery (89.5%). The six patients where PAP prescription was prolonged for more than 24 h received PAP for an average of two days (mean 2.00; SD 1.67).

Route of administration: All patients had the appropriate route of administration (100%).

Redosing: Redosing was appropriate for 81 of 83 patients (95.7%). There were no patients with blood loss over 1500 mL and two patients with a surgical duration over 3 h, which requires a second dose according to the local guidelines. However, neither of these two patients received an extra dose of antibiotics.

Univariate logistic regression analysis was performed for QIs 1 (correct indication), 2 (correct agent), 4 (correct timing), and cumulative appropriateness, which all scored less than 90% appropriateness. For QI1, three variables (BMI, duration of procedure and duration of preoperative stay), for QI2, two variables (age and knowledge of renal function), for QI4, two variables (age and AZV), for QI5, two variables (AZV and preoperative antibiotic use), and for cumulative compliance, three variables (age, BMI and duration of preoperative stay) were selected and entered into the multivariable logistic regression analysis.

The multivariable logistic regression analysis demonstrated that older patients more often received an appropriate agent (odds ratio (OR) per ten years increase in age = 0.52; CI95% 0.31–0.87; *p* = 0.013), but more often fell to inappropriate timing (OR = 1.79; CI95% 1.09–2.92; *p* = 0.020). Patients who received an implant had an appropriate duration more often (OR = 0.11; CI95% 0.02–0.81). The multivariable logistic regression analysis demonstrated that no variables were significantly associated with QI1 (correct indication) and cumulative appropriateness. The results of the univariate and multivariable logistic regression analysis can be seen in Supplementary Appendix A.

The kappa value for interrater reliability for the registration of time of administration noted by the researchers and by OT staff was 0.039 and the kappa value for the registration of time of first incision stated by the researchers and by the OT staff was 0.197. The kappa for interrater reliability between the two ID physicians was 0.969.

### 2.2. Interviews

In total, eleven professionals involved in PAP were interviewed. Two anaesthesiologists (of whom one temporary), three nurse anaesthetists and one OT nurse volunteered to participate in face-to-face interviews. One general surgeon, one orthopaedic surgeon, one ID physician, one gynaecologist and one pharmacist were interviewed via videoconferencing. Of all interviewees, seven were male and four were female. One surgeon was trained in Latin America and the rest of the interviewees in Western Europe. Interviews lasted between seven and 35 min. Ten interviews were conducted in Dutch and one in English.

A deductive framework analysis revealed guideline factors and hospital factors as the main themes influencing PAP guideline adherence. Guideline factors are barriers associated with the guidelines. The sub-themes that were found in our interviews were knowledge, attitudes towards the guidelines and the practicality of the guidelines. Hospital factors were barriers associated with HOH organizational culture. The sub-themes identified were professional interactions, capacity for change and resources.

Most of the interviewees reflected on their awareness and knowledge of the guidelines. Most mentioned barriers were unclarity about antibiotic allergies and poor knowledge of surgeons about the guidelines content itself. Moreover, the interviews made clear that the attitude towards the PAP guidelines was generally negative; most professionals do not use the local guidelines because they use the guidelines of their own specialism federation instead or PAP is considered as least important responsibility when performing patient preparation on the OT. Finally, barriers related to the practicality of the guidelines were mentioned by all participants. Most of them suggested that the accessibility of the guidelines was difficult, that the compatibility of the guidelines was questionable due to OT planning and available infusion systems and that the guidelines were not completely culturally appropriate. Indicative quotations are presented in Table 3.

Regarding hospital factors, interviewees mentioned no clearly defined responsibilities, lack of explanation of the guidelines to new (temporary) employees, negative feelings of hierarchy and multiple (unnecessary) referral moments, as most the important barriers for PAP guideline adherence concerning professional interactions. In addition, the interviews implied that the HOH had a lot of rules, which was experienced as a barrier for change and the improvement of care, and thus better guideline adherence. Finally, most of the interviewees discussed the lack of resources as an important theme in the HOH. Most mentioned barriers in this sub-theme were limited checkpoints for PAP administration, usage of old information systems and unavailability of different antimicrobial agents on the OT when substitute PAP is indicated. Indicative quotations are presented in Table 4.

### 2.3. Survey

The survey contained questions about the most mentioned barriers from the interviews, namely guideline barriers, appropriate time administration, adherence, PAP allergies, professional growth, responsibilities, communication, authority, resources and capacity for change. It was sent to 37 recipients, of which 24 (64.9%) completed the survey. The respondents included five anaesthesiologists, four nurse anaesthetists, one temporary nurse anaesthetist, twelve surgeons, one ID physician and one pharmacist. The mean age was 46.4 years (SD = 9.32) and median months of employment was 72 months (range 5–240). Among the respondents, 66.7% (*n* = 16) were male and the majority of the respondents was educated in the Netherlands (70.8%, *n* = 17). Figure 1 shows the results of the statements.

Most important barriers revealed in the survey (Figure 2) were a lack of explanation of the guidelines to new (temporary) employees and a deficient OT planning resulting in the inappropriate timing of PAP administration. Respondents thought that routine prescribing behaviour by surgeons resulted in more PAP administration than dictated by the guidelines. In addition, respondents believed little knowledge was available about the recommendations in case of allergies, and unclarity existed about responsibilities with regard to the choice of type of PAP. Finally, the communication with the AST was reported to be poor. Most suggested solutions for these barriers were to provide more feedback and more education about postoperative infections and/or complications and resistance data via clinical lessons, newsletters and specialist presentations.

## 3. Discussion

Little is known about barriers to appropriate PAP prescription in SIDS. This study was the first to evaluate the appropriateness and to identify barriers for guideline adherence of PAP in Aruba. Overall, the scores for the quality indicators on the correct indication, the correct choice of antimicrobial agent and the correct timing of PAP were poor. The overall appropriateness of PAP was found to be only 34.9%. The main barriers reported were poor knowledge about PAP and the PAP guidelines as well as poor professional interactions regarding PAP.

Even though this study used similar QIs compared to previous research [16], the outcomes differ. Large variability in appropriateness was found in the literature, both globally as for each criterion (Table 5). This variability is possibly attributable to differences in study populations, methodologies and criteria for appropriateness. Therefore, comparisons with other studies should be made with caution. In addition, results can differ from previous research because it is difficult to implement international guidelines in SIDS due to organizational differences [17].

The multivariable logistic regression analysis demonstrated that older patients more often received an appropriate agent, but more often had an inappropriate timing. Larger studies are needed to show whether these associations are due to specific factors, such as the type of procedure. Our study showed that patients with an implant more often had an appropriate duration. However, the CI95% was wide, so nothing concrete can be concluded and therefore subsequent studies are necessary. We did not find any association with the QIs correct indication and cumulative appropriateness. Previous research found similar [19] and other associations [19,24].

The interrater reliability for the administration times and incision times noted by the researchers compared to the OT staff was low, resulting in a different score for the QI correct timing. The OT staff is a heterogeneous group, so we cannot conclude that every single professional registrates poorly. Nevertheless, since the kappa is low, it is important to educate the professionals about accurate registration. This will make future audits using patient files and OT forms more reliable. An electronic registration system might help to make the registration easier and more accurate, as mentioned by the interviewees.

Our study demonstrated that the scores of the QIs correct indication, agent and timing were the lowest. Routine prescribing behaviour by surgeons appeared to be a possible reason for the lower score of correct indication and correct agent, as also demonstrated in previous research [25]. Poor knowledge can be an explanation for routine prescribing behaviour and therefore high antibiotic prescription [26]. A reason that might explain the lower scores for the QI correct timing is that the local PAP guidelines recommend a different time frame (15–60 min before incision) [27] than the WHO (0–120 min before incision) [28]. When the criteria of the WHO were used in this study, the score for correct timing would be 95% instead of 55%. Therefore, the local guidelines should be critically revised.

In addition to specific barriers for the QIs correct indication, agent and timing, we identified other aspects that might have negatively influenced all the QI scores. Our study identified poor knowledge as a barrier to guideline adherence, which was also identified in previous studies [25,29,30,31]. This could be a result of a lack of education and feedback. Moreover, our results show that poor communication negatively influences guideline adherence. Negative feelings of hierarchy and the multicultural composition of OT teams with different educational backgrounds, as is often the case in SIDS, are the main causes. Improving communication can lead to better guideline adherence. [32,33]. Poor communication can also result in unclarities about responsibilities, because PAP responsibilities are not delegated [25]. This can result in a poor continuity of care [32,34,35,36]. Unclarities about responsibilities and a poor continuity of care were also found in our study as barriers.

Our study results in several recommendations for Aruba that could also be applicable to other SIDS to improve PAP guideline adherence and therefore enhance patient care. Our main recommendations are focused on education, communication and awareness of the possibility to consult an ID physician or microbiologist.

We advise to implement clinical lessons and feedback moments for all OT professionals to improve knowledge. Improving knowledge positively influences the attitude towards the local guidelines [37] and PAP. It is important to provide these clinical lessons and feedback moments regularly, because of the frequent changing staff composition. The AST could educate the OT staff, with a focus on PAP, the importance of guideline adherence and AMR. Education should also focus on communication skills [38], so negative feelings of hierarchy and unclarities about responsibilities will reduce. We recommend that SIDS should collaborate on the training of (young) professionals, sharing educational materials and sharing data about AMR and PAP use. This recommendation is a form of capacity-building, which is common in SIDS.

In Aruba, and possibly other SIDS, OT professionals are unaware of the availability to consult an ID physician or microbiologist, if these are even available in the hospital. We recommend increasing the awareness of the possibility to consult ID physicians or microbiologists in case of unclarities, which is likely to reduce inappropriate PAP choices. When these specialists are not available, we recommend that hospitals collaborate to establish a possibility to online consult such a specialist from another hospital. Sharing knowledge and trained staff among SIDS will help them build capacity.

This study has several strengths. It was the first study that evaluated the appropriateness and barriers of PAP guideline adherence in Aruba, a SIDS, both quantitatively and qualitatively. Findings from the audit, interviews and survey has resulted in a true mixed-methods approach that is state-of-the-art in quality-of-care research. The use of verbatim transcripts and the coding approach improved the analytical rigor and validity of the interview analysis. The interrater reliability was very good between the two independent Dutch and Aruban ID physicians, who assessed the correct indication, agent and dose, which indicates an accurate assessment of these QIs based on the local guidelines.

This study also has some limitations. The COVID-19 pandemic resulted in a shorter study period and has consequently limited patient recruitment. This might have resulted in a questionable value of the multivariable analysis. Unfortunately, this small sample size was unavoidable. Only adherence to the PAP guidelines was studied and not to other guidelines on antibiotic therapy, making it unclear whether these guidelines show better compliance. However, since this study focused on the PAP guidelines only, opinions about barriers for PAP guideline adherence are unequivocal. One interview was conducted in English while the other interviews were performed in Dutch, which could have influenced the self-expression of the researchers and interviewees. However, this was unavoidable.

Despite these limitations, these data are of importance because it provides information that contributes to the understanding of the appropriateness and the barriers of PAP guideline adherence in Aruba. Since Aruba is part of the Caribbean SIDS, these study results are interesting for other SIDS.

## 4. Materials and Methods

### 4.1. Setting

The HOH in Aruba is a medium-sized 172-bed hospital offering all major specialties and all relevant surgical procedures. Aruba is a small Caribbean island with approximately 200,000 residents. The multicultural composition of OT teams of Aruba results in specialists with different (educational) backgrounds, among others Dutch, Latin American and Northern American. The HOH uses local PAP guidelines [27] which are based on Dutch SWAB Guidelines [39].

### 4.2. Ethical Approval

Ethical approval was allocated by the Medical Ethical Committee of the HOH. For the audit, informed consent was not obtained from patients because no intervention at patient level was present and patient data were analysed anonymously aiming at improving the quality of healthcare. For the interviews, written informed consent was obtained at the beginning of the interview.

### 4.3. Audit

The first part of this study was a prospective audit of the appropriateness of PAP. All patients ≥18 years old undergoing surgery between 5 March 2020 and 12 March 2020 were enrolled.

For each patient, the following data were collected in case report forms: (i) patient characteristics including age, sex, antibiotic allergies, duration of preoperative hospital stay, weight, height, BMI, current use of antibiotics, ESBL status, MRSA status, renal function (GFR), American Society of Anaesthesiologists physical status (ASA) score and AZV insurance (Aruban health insurance, to distinguish between locals and non-locals); (ii) treatment data including procedure, surgical ward ((ENT (ear, nose, throat), general surgery, orthopaedic surgery, gynaecology surgery, plastic surgery, cardiac device implants, urological surgery and neurosurgery), date of procedure, use of implant (cardiac devices or orthopaedic prosthetics), duration of procedure (minutes), type of procedure (elective, emergency (within 24 h) or acute (within 3 h)), anaesthesiologist and (head) surgeon; and (iii) PAP data including antimicrobial agent, dosage, administration route, time of administration, time of incision or tourniquet inflation, duration and consultation with an infectious diseases (ID) physician.

The data of all patients were collected from patient files, OT reports and medication charts, documented by OT staff and additional PAP data were collected via researchers’ observations (SB, LvM). Data were entered into Castor EDC (Castor EDC, Amsterdam, the Netherlands: Castor). To reduce the Hawthorne effect, the operating staff were unaware of the assessed criteria.

#### 4.3.1. Quality Indicators

The appropriateness of PAP was assessed for seven quality indicators (QIs), based on the quality indicators defined by Monnier et al. [16], namely correct (i) indication, (ii) antimicrobial agent, (iii) dose, (iv) timing, (v) duration, (vi) route of administration and (vii) redosing. Algorithms were developed for every QI, using a syntax in Microsoft Excel (Microsoft Excel 2017, Redmond, Washington: Microsoft). Furthermore, cumulative appropriateness was calculated for all Is. The local guidelines [27] were used to set criteria for the compliance of the QIs and thereby assess appropriateness per QI. Correct indication, antimicrobial agent and dose were assessed for every surgical procedure by two independent Dutch and Aruban ID physicians (JdV, JdK) based on the description of indications in the prevailing HOH antibiotic guidelines [27].

#### 4.3.2. Statistical Analysis

Statistical analysis was conducted using IBM SPSS Statistics (IBM Corp. Released 2019. IBM SPSS Statistics for Macintosh, Version 26.0. Armonk, NY: IBM Corp). Descriptive statistics were derived using frequencies, mean, standard deviation (SD), median and range values. All variables were tested in a univariate logistic regression model. When a *p*-value ≤ 0.25 was obtained, the variable was entered in a multivariable logistic regression model [19]. If the model did not allow for more variables to be added, the researchers chose via discussion the variables that were most likely to have clinical value.

Cohen’s kappa coefficient was used to determine the interrater reliability between the independent ID physicians who set the indication of PAP and assessed the correctness of the prescribed PA (JdV, JdK), and the administration and incision times observed by the researchers and OT staff.

### 4.4. Interviews

The second part of the study, conducted alongside the audit, consisted of interviews to identify barriers for PAP guideline adherence, followed by a survey to verify the outcomes of the interviews.

OT professionals were randomly selected with stratification based on department, to achieve a variation in influencing factors regarding PAP decision making [40]. Interviews were performed until data saturation was reached. Professionals were invited to participate by explaining participant information orally or via email and providing the informed consent form.

Interviews were executed using a semi-structured protocol developed via review of literature and discussion with other researchers (JS, JdK, JdV, SB). Additionally, the interview questions were based on the ‘Tailored Implementation for Chronic Diseases’ (TICD) checklist developed by Flottorp et al. [15]. Interviews were always performed by two researchers (LvM, SB). Respondent bias was reduced with the use of open questions, gaining trust from interviewee to interviewers and asking more in detail when dishonest or incorrect answers were expected. The interviews were transcribed verbatim and anonymized. Moreover, 5% of the transcripts were compared by a second researcher (SB) to check the transcription accuracy.

Deductive coding of transcripts was executed with pre-existing codes based on the interview protocol, using Atlas.ti 8 (ATLAS.ti 8, Berlin, Germany: Scientific Software Development GmbH). To improve the internal validity and interrater reliability, coding was independently performed by two researchers (LvM, SB), with cross-checking and discussion of the coding to gain a complete interpretation of the data.

Analysis of the data was performed using a framework approach, including the following phases [41]:Familiarizing with the data;Generating initial codes;Searching for themes;Reviewing themes;Defining and naming themes;Final analysis and producing the results.

Analytical rigor was increased by searching for negative, atypical and conflicting or contradicting cases in coding and themes. Personal experiences of the researcher, which may influence the analysis, were described in reflective memos. Furthermore, the interpretations of the interviews were discussed with other researchers (J.d.V., J.d.K., J.S.) to increase the validity of the results [42].

### 4.5. Survey

A quantitative survey was electronically distributed amongst all OT professionals (37) who were not interviewed on 9 May 2020, and a reminder was sent after seven days. The survey contained demographics and was developed after searching for patterns in the interviews and discussion with the research team. The survey contained 32 to 35 items, depending on answers of the respondent, with 20 to 23 multiple choice questions and 12 statements, which employed a five-level Likert-type rating scale [43]. Completion of the survey was voluntary and informed consent was provided at the start of the survey.

Descriptive statistics were derived about respondent characteristics and responses to the questions and statements using Microsoft Excel software. Responses of statements were scored as completely agree (5); agree (4); not disagree/not agree (3); disagree (2); completely disagree (1); and I do not know/not applicable (0). Weighted means were calculated and disagreement was defined as <3 points, agreement was defined as 3 points and strong agreement was defined as ≥4 points [44].

## 5. Conclusions

Adherence to several QIs relevant for PAP were found to be low in this study, namely correct indication, correct antimicrobial agent and correct timing. The most important barriers for PAP guideline adherence were poor knowledge and poor professional interactions, most likely due to lack of education, multicultural composition of OT teams, poor communication and unclarity about responsibilities. Furthermore, accurate registration was found to be difficult in small island setting even if it is essential to perform reliable audits. Results from this study are likely to reflect PAP practices in other small islands developing states, with implications particularly for the Caribbean nations. Therefore, we believe it is important for these settings to focus on communication, education and the awareness of the possibility to consult an ID physician or microbiologist to improve PAP guideline adherence.

## Figures and Tables

**Figure 1 antibiotics-10-00462-f001:**
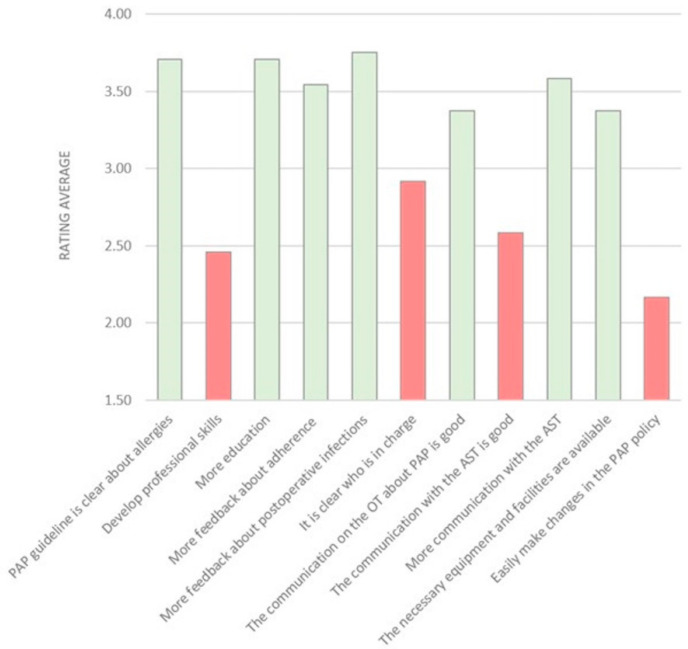
Rating averages of the statements in the survey. Red represents disagreement about the statement and green represents agreement about the statement. N respondents is 24.

**Figure 2 antibiotics-10-00462-f002:**
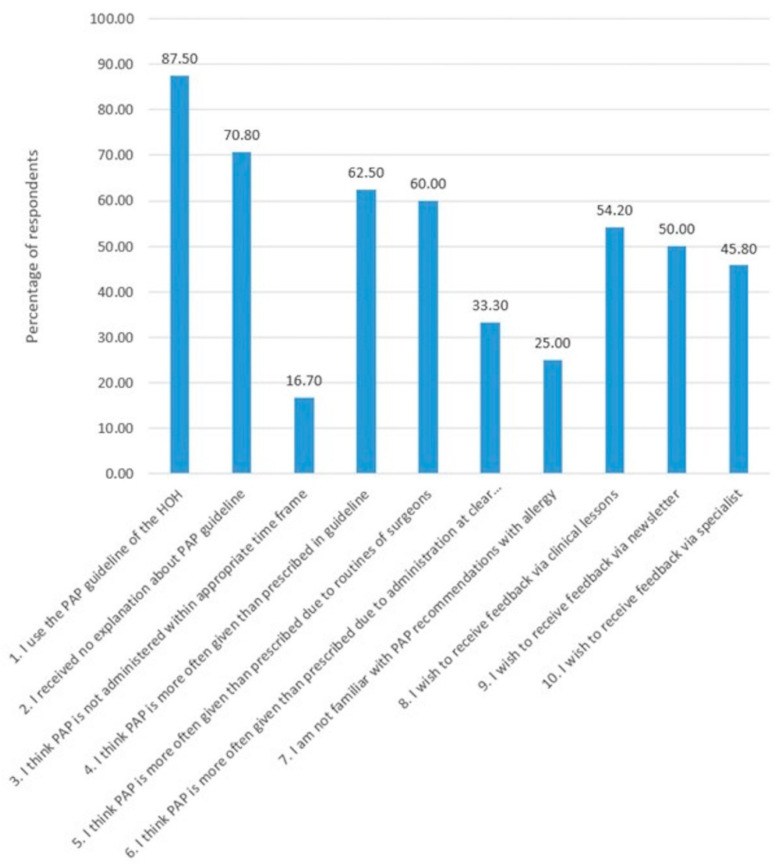
Barriers for PAP guideline adherence represented for the percentages of respondents. Questions 5, 6 and 7 were subquestions when respondents answered question 4 with “yes”, so these were answered by 15 respondents. The remaining questions were answered by 24 respondents.

**Table 1 antibiotics-10-00462-t001:** Patients’ characteristics and surgical information.

Characteristic	*n* (%)	Characteristic	*n* (%)
Surgical procedures	83 (100)	Implant	11 (13.3)
Age (mean; SD)	54.1;17.7	Elective procedure	74 (89.2)
Female	57 (68.7)	Emergency procedure	8 (9.6)
AZV insurance	77 (92.8)	Acute procedure	1 (1.2)
ESBL positive	1 (1.2)	ENT surgery	3 (3.6)
BMI (mean; SD)	31.9; 6.6	Neurosurgery	10 (12)
Allergy to penicillin	3 (3.6)	Orthopaedic surgery	10 (12)
ASA score I	12 (14.5)	Gynaecology surgery	19 (22.9)
ASA score II	41 (49.4)	General surgery	23 (27.7)
ASA score III	23 (27.7)	Urological surgery	5 (6.0)
ASA score IV	2 (2.4)	Cardiac device implants	6 (7.2)
ASA score missing	5 (6)	Plastic surgery	7 (8.4)

**Table 2 antibiotics-10-00462-t002:** Appropriateness of perioperative prophylaxis per quality indicator and cumulative compliance.

Quality Indicator	*n* (%)
QI1: Compliance to indication	42 (50.6)
QI2: Appropriate agent	16 (30.8)
QI3: Appropriate dose	51 (94.4)
QI4: Appropriate timing based on observations of researchers	22 (55.0)
QI4: Appropriate timing based on reported incision and administration times	21 (42.9)
QI5: Appropriate duration	51 (89.5)
QI6: Appropriate route of administration	55 (100)
QI7: Appropriate redosing after 3 h or >1500 mL blood loss	44 (95.7)
Cumulative compliance	29 (34.9)

**Table 3 antibiotics-10-00462-t003:** Guideline factors: Indicative quotations.

Participant	Subtheme	Indicative Quotation
S12 Gynaecologist	Practicality	I think it is easy to follow, it is Dutch, so I have to translate it because I am just learning Dutch. Yeah, it’s understandable. Doesn’t make any difficulties. Maybe sometimes they use a lot of abbreviations, but eh I don’t know what these abbreviations mean.
M2 Nurse anaesthetist	Attitudes towards the guidelines	I have no idea. I’m not really sure what to think. You know, some surgeons are very spastic about it, like eh cardiologists are really spastic about that it must be eh half an hour to three quarters before incision, and he doesn’t want longer, and shorter is also not possible, then I think “yes, that can just not be organized here”.
S23 General surgeon	Attitudes towards the guidelines	An example is that when we have done the time out procedure, we mentioned the antibiotics, dose and time when it is done, and 10 min after the start of the surgery, someone checks it once again and they say “was the antibiotic prophylaxis given or not?”, so no one paid attention.
I1 ID physician	Knowledge	In the 80, 90 percent that surgeons just have to give cefazolin, it is just fine. But everything different, if patients have allergies, or are MRSA positive, et cetera, then their knowledge will stop very quickly. And then they do not actively ask us, because that is also possible via the ID physician telephone consultations, which is manned by one of us Monday to and Friday, so if they have questions, they do not do that via the ID physician telephone consultations, and then they usually just give cefazolin, or they invent something themselves.
S23 General surgeon	Knowledge	Clarity to everyone, to the staff, to the OT. It’s what I said, it’s kind of a mess. One, they don’t even know there was a protocol, eh let alone that people adhere to the protocol.

**Table 4 antibiotics-10-00462-t004:** Hospital factors: indicative quotations.

Participant	Subtheme	Indicative Quotation
M2 Nurse anaesthetist	Professional interactions	Oh, well you know, addressing is not the problem, I do that. But yes, you know, they’re the surgeon and if he has a very good reason for that. I don’t have the medical background to have an opinion about that. That’s not for me, it’s not that I don’t dare, but it’s not for me to talk about that. It’s not my specialization, and for them it is “very important what you think, but I will do it anyway”.
S23 General surgeon	Professional interactions	That [admission] form states antibiotic prophylaxis, yes or no, if yes, according to protocol question mark. But I don’t write which [type of PAP] on my admission form, because the admission form gives the opportunity to do it according to protocol. So I always fill in protocol.
M3 Nurse anaesthetist	Resources	Before that hack took place, we had Chipsoft. It was our responsibility [to register PAP], but actually the operation theatre nurses mainly did that, who then asked to us, “how long was the antibiotics in it”, and if uhm, they wrote of the amount, the type of antibiotics and the time it was administered. Uh, but we don’t have that now, we have the registration list as replacement for Chipsoft. Officially, uhm, we have to write down the time of the antibiotics, but that doesn’t happen. So actually, all we have left in terms of registration is the anaesthesia list.
M2 Nurse anaesthetist	Capacity for change	Well, I guess I’m kind of used to that <chuckles>. After seven years, at a certain moment, you know that nothing is going to change. So I’m not going to put my energy into it.
P1 Pharmacist	Capacity for change	But usually when new guidelines are introduced or something like that… No, the hospital does not regulate anything, you have to do it all yourself.

**Table 5 antibiotics-10-00462-t005:** Compliance rates to local guidelines in multiple countries worldwide.

Study	Country	Indication	Agent	Dose	Timing	Duration	Route of Administration	Redosing	Overall Compliance
Current study	Aruba	50.6%	30.8%	94.4%	55.0%	89.5%	100%	95.7%	34.9%
Quattrocchi et al. (2018) (hospital A and B) [18]	Italy	A: 72.3% B: 77.9%	A: 87.8% B: 9.8%		A: 89.1% B: 78.4%	A: 99.0% B: 8.9%			A: 40.7% B: 0.8%
Napolitano et al. (2013) [19]	Italy	81.4%	25.5%						18.1%
Alahmadi et al. (2020) [20]	Saudi Arabia				22.5%	56.4%			19.5%
Khan et al. (2020) [21]	Pakistan		4.2%		51%		100%		
Koek et al. (2017) [22]	The Nether-lands								85%
Hohmann et al. (2012) [23]	Germany					67.1%			70.7%

## Data Availability

The data presented in this study are available on request from the corresponding author. The data are not publicly available due to privacy.

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
