# Peer review of "Barriers and Facilitators in Perioperative Antibiotic Prophylaxis: A Mixed-Methods Study in a Small Island Setting"

_antibiotics, 2021, doi:10.3390/antibiotics10040462_

Round 1
Reviewer 1 Report
I read with great interest the manuscript. I find it well wrote and with good idea research.
Below my suggestions
- Introduction: updata data on AMR wordwilde. About surgical site infection are the cause also for high maternal mortality in Africa. see and cite this paper from Sierra Leone (Maternal caesarean section infection (MACSI) in Sierra Leone: a case-control study. Epidemiol Infect. 2020 Feb 27;148:e40.)
- Methods section: clear
- Results: are well presented
- Discussion: well discuss also the role of young doctors and education during medical school in Antimicrobial resistance. In fact training young people means planning for the future. AMR should be a priority for politicians and for all health workers. The inclusion of competencies in antibiotic use in all specialty curricular is urgently needed
- Furthermore, add some public health action that came from your interesting paper
Author Response
Point 1: Introduction: update data on AMR worldwide. About surgical site infection are the cause also for high maternal mortality in Africa. see and cite this paper from Sierra Leone (Maternal caesarean section infection (MACSI) in Sierra Leone: a case-control study. Epidemiol Infect. 2020 Feb 27;148:e40.)
Response 1: Thank you for bringing this paper to our attention, we indeed believe this should be part of the introduction. Therefore, we added the high maternal mortality in Sierra Leone as an concrete example of a consequence of SSIs in lines 36 and 37 and referred to the proposed article. Additionally, we added information on antimicrobial resistance worldwide in lines 42 and 43 by referring to the World Health Organisation statement.
Point 2: Discussion: well discuss also the role of young doctors and education during medical school in Antimicrobial resistance. In fact training young people means planning for the future. AMR should be a priority for politicians and for all health workers. The inclusion of competencies in antibiotic use in all specialty curricular is urgently needed
Response 2: Thank you for your comment. We also believe it is highly important to focus on the role of young doctors and education during medical school in the prevention of antimicrobial resistance. Therefore, we added the aspect of training (young) professionals in line 259 till 262. We chose not to further elaborate on this aspect, since the role of young doctors and education during medical school was not part of our study outcomes. However, we agree that the inclusion of competencies in antibiotic use is definitely important, which is the reason we recommend implementing clinical lessons with a focus on PAP, guideline adherence, and AMR. We added the latter in line 257 and 258.
Point 3: Furthermore, add some public health action that came from your interesting paper
Response 3: Thank you for this suggestion. We agree that public health action is very important and also recognized by the WHO as an important aspect in healthcare. However, our study focused on in hospital antibiotic use, in which improving skills of the operation theater staff is of great importance. Public health action is undertaken but has no direct relation with the subject of our study. Therefore, we emphasized capacity-building in our recommendation in lines 259 till 262.

Reviewer 2 Report
Major comments:
-short study perid and limited number of patients are the limitations of the study, but have already been mentioned in the chapter discussion.
Minor commets:
-line 56 ,the acronym HOH is mentioned the first time but is not explained
(Dr.Horacio E.Oduber Hospital)
-The acronym OR has two meanings; Odds ratio and operating room. Probably the authors change the acronym for operating room (OPR?)
Author Response
Point 1: Short study period and limited number of patients are the limitations of the study, but have already been mentioned in the chapter discussion.
Response 1: Unfortunately, we were not able to include more patients in our study or increase the study period due to the Covid-19 pandemic, as explained in our limitations. In a further study more participants should be enrolled in a larger period of time.
Point 2: Line 56, the acronym HOH is mentioned the first time but is not explained.
Response 2: Thank you for your comment. In our revision, we explain the acronym HOH in lines 58 and 59.
Point 3: The acronym OR has two meanings; Odds ratio and operating room. Probably the authors change the acronym for operating room (OPR?).
Response 3: Thank you for your attention on this point. We changed the term operating room into operating theatre, and therefore the acronym OR in OT throughout the whole manuscript. Additionally, we changed the acronym in OT in figure 1.
